# Factor Graph Grammars

**David Chiang**
University of Notre Dame
dchiang@nd.edu

**Darcey Riley**
University of Notre Dame
darcey.riley@nd.edu

## Abstract

We propose the use of hyperedge replacement graph grammars for factor graphs, or *factor graph grammars* (FGGs) for short. FGGs generate sets of factor graphs and can describe a more general class of models than plate notation, dynamic graphical models, case–factor diagrams, and sum–product networks can. Moreover, inference can be done on FGGs without enumerating all the generated factor graphs. For finite variable domains (but possibly infinite sets of graphs), a generalization of variable elimination to FGGs allows exact and tractable inference in many situations. For finite sets of graphs (but possibly infinite variable domains), a FGG can be converted to a single factor graph amenable to standard inference techniques.

## 1   Introduction

Graphs have been used with great success as representations of probability models, both Bayesian and Markov networks (Koller and Friedman, 2009) as well as latent-variable neural networks (Schulman et al., 2015). But in many applications, especially in speech and language processing, a fixed graph is not sufficient. The graph may have substructures that repeat a variable number of times: for example, a hidden Markov model (HMM) depends on the number of words in the string. Or, part of the graph may have several alternatives with different structures: for example, a probabilistic context-free grammar (PCFG) contains many trees for a given string.

Several formalisms have been proposed to fill this need. Plate notation (Buntine, 1994), plated factor graphs (Obermeyer et al., 2019), and dynamic graphical models (Bilmes, 2010) address the repeated-substructure problem, but only for sequence models like HMMs. Case–factor diagrams (McAllester et al., 2008) and sum–product networks (Poon and Domingos, 2011) address the alternative-substructure problem, so they can describe PCFGs, but only for fixed-length inputs.

More general formalisms like probabilistic relational models (Getoor et al., 2007) and probabilistic programming languages (van de Meent et al., 2018) address both problems successfully, but because of their generality, tractable exact inference in them is often not possible.

Here, we explore the use of *hyperedge replacement graph grammars* (HRGs), a formalism for defining sets of graphs (Bauderon and Courcelle, 1987; Habel and Kreowski, 1987; Drewes et al., 1997). We show that HRGs for factor graphs, or factor graph grammars (FGGs) for short, are expressive enough to solve both the repeated-substructure and alternative-substructure problems, and constrained enough allow exact and tractable inference in many situations. We make three main contributions:

- We define FGGs and show how they generalize the constrained formalisms mentioned above (§3).

- We define a *conjunction* operation that enables one to modularize a FGG into two parts, one which defines the model and one which defines a query (§4).

- We show how to perform inference on FGGs without enumerating the (possibly infinite) set of graphs they generate. For finite variable domains, we generalize variable elimination to FGGs

(§5.1). For some FGGs, this is exact and tractable; for others, it gives a sequence of successive approximations.

For infinite variable domains, we show that if a FGG generates a finite set, it can be converted to a single factor graph, to which standard graphical model inference methods can be applied (§5.2). But if a FGG generates an infinite set, inference is undecidable (§5.3).

## 2 Background

In this section, we provide some background definitions for hypergraphs (§2.1), factor graphs (§2.2), and HRGs (§2.3). Our definitions are mostly standard, but not entirely; readers already familiar with these concepts may skip these subsections and refer back to them as needed.

### 2.1 Hypergraphs

Assume throughout this paper the following "global" structures. Let $L^V$ be a finite set of *node labels* and $L^E$ be a finite set of *edge labels*, and assume there is a function $type : L^E \to (L^V)^*$, which says for each edge label what the number and labels of the endpoint nodes must be.

**Definition 1.** A *hypergraph* (or simply a *graph*) is a tuple $(V, E, att, lab^V, lab^E)$, where

- $V$ is a finite set of *nodes*.

- $E$ is a finite set of *hyperedges* (or simply *edges*).

- $att : E \to V^*$ maps each edge to zero or more *endpoint* nodes, not necessarily distinct.

- $lab^V : V \to L^V$ assigns labels to nodes.

- $lab^E : E \to L^E$ assigns labels to edges.

- For all $e$, $|att(e)| = |type(lab^E(e))|$, and if $att(e) = v_1 \cdots v_k$ and $type(lab^E(e)) = \ell_1 \cdots \ell_k$, then $lab^V(v_i) = \ell_i$ for $i = 1, \ldots, k$.

Although the elements of $V$ and $E$ can be anything, we assume in our examples that they are natural numbers. If a node $v$ has label $\ell$, we draw it as a circle with $\ell_v$ inside it. We draw a hyperedge as a square with lines to its endpoints. In principle, we would need to indicate the ordering of the endpoints somehow, but we omit this to reduce clutter.

### 2.2 Factor graphs

**Definition 2.** A *factor graph* (Kschischang et al., 2001) is a hypergraph $(V, E, att, lab^V, lab^E)$ together with mappings $\Omega$ and $F$, where

- $\Omega$ maps node labels to sets of possible values. For brevity, we write $\Omega(v)$ for $\Omega(lab^V(v))$.

- $F$ maps edge labels to functions. For brevity, we write $F(e)$ for $F(lab^E(e))$. For every edge $e$ with $att(e) = v_1 \cdots v_k$, $F(e)$ is of type $\Omega(v_1) \times \cdots \times \Omega(v_k) \to \mathbb{R}_{\geq 0}$.

A node $v$ together with its domain $\Omega(v)$ is called a *variable*. An edge $e$ together with its function $F(e)$ is called a *factor*.

We draw a factor $e$ as a small square, but instead of writing its label, we write $F(e)$ next to it, as an expression in terms of its endpoints. As shorthand, we often write Boolean expressions, which are implicitly converted to real numbers (true = 1 and false = 0).

**Example 3.** Although HMMs are defined for sentences of arbitrary length, factor graphs force us to choose a fixed length; below is a HMM for sentences of length 3. (Here, $\mathbf{T}$ and $\mathbf{W}$ are node labels, $\Omega(\mathbf{T})$ is the set of possible tags, and $\Omega(\mathbf{W})$ is the set of possible words.)

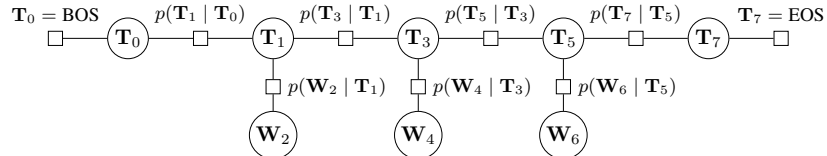

**Definition 4.** If $H$ is a factor graph, define an *assignment* $\xi$ of $H$ to be a mapping from nodes to values: $\xi(v) \in \Omega(v)$. We write $\Xi_H$ for the set of all assignments of $H$. The *weight* of an assignment $\xi$ is given by

$$w_H(\xi) = \prod_{\substack{\text{edges } e \\ \text{with } att(e) = v_1 \cdots v_k}} F(e)(\xi(v_1), \ldots, \xi(v_k)).$$

In a factor graph with no factors, every assignment has weight 1. A factor graph with no variables has exactly one assignment.

Factor graphs are general enough to represent Bayesian networks and Markov networks. They can also represent stochastic computation graphs (SCGs), introduced by Schulman et al. (2015) for latent-variable neural networks.

## 2.3 Hyperedge Replacement Graph Grammars

Hyperedge replacement graph grammars (HRGs) were introduced by Bauderon and Courcelle (1987) and Habel and Kreowski (1987), and surveyed by Drewes et al. (1997). They generate graphs by using a context-free rewriting mechanism that replaces nonterminal-labeled edges with graphs. In this section, we provide a brief definition of HRGs, with a minor extension for node labels.

**Definition 5.** A *hypergraph fragment* is a tuple $(V, E, att, lab^V, lab^E, ext)$, where

- $(V, E, att, lab^V, lab^E)$ is a hypergraph,
- $ext \in V^*$ is a sequence of zero or more *external nodes*.

In our figures, we draw external nodes as black nodes. In principle, we would need to indicate their ordering somehow, but we omit this to reduce clutter.

**Definition 6.** A *hyperedge replacement graph grammar* (HRG) is a tuple $(N, T, P, S)$, where

- $N \subseteq L^E$ is a finite set of *nonterminal symbols*.
- $T \subseteq L^E$ is a finite set of *terminal symbols*, such that $N \cap T = \emptyset$.
- $P$ is a finite set of *rules* of the form $(X \to R)$, where
  - $X \in N$.
  - $R$ is a hypergraph fragment with edge labels in $N \cup T$.
  - If $R$ has external nodes $x_1 \cdots x_k$, then $type(X) = lab^V(x_1) \cdots lab^V(x_k)$.
- $S \in N$ is a distinguished *start nonterminal symbol* with $type(S) = \epsilon$.

Although a left-hand side $X$ is formally just a nonterminal symbol, we draw it as a hyperedge labeled $X$ inside, with replicas of the external nodes as its endpoints. On right-hand sides, we draw an edge $e$ with nonterminal label $X$ as a square with $X_e$ inside. If $R$ is the empty graph, we write $\emptyset$.

Intuitively, a HRG generates graphs by starting with a hyperedge labeled $S$ and repeatedly selecting an edge $e$ labeled $X$ and a rule $X \to R$ and replacing $e$ with $R$. (See Figure 1a for an example, where $H = R$.) Replacement stops when there are no more nonterminal-labeled edges.

As with a CFG, we can abstract away from the ordering of replacement steps using a *derivation tree*, in which the nodes are labeled with HRG rules, and an edge from parent $\pi_1$ to child $\pi_2$ has a label indicating which edge in the right-hand side of $\pi_1$ is replaced with the right-hand side of $\pi_2$.

**Definition 7.** Let $G$ be a HRG. For all nonterminals $X$, define the set $\mathcal{D}(G, X)$ of $X$-*type derivation trees* (or simply $X$-*type derivations*) of $G$ to be the smallest set containing all finite, unordered, edge-labeled trees of the form shown in Figure 1b, where $\pi = (X \to R)$ is a rule in $G$, $R$ has nonterminal-labeled edges $e_1, \ldots, e_k$ with labels $X_1, \ldots, X_k$, and for $i = 1, \ldots, k$, $D_i$ is an $X_i$-type derivation. We simply write *derivation* for $S$-type derivation, and we let $\mathcal{D}(G) = \mathcal{D}(G, S)$.

The *derived graph* of a derivation $D$ is the graph formed as follows. If $D$ is as shown in Figure 1b, then for $i = 1, \ldots k$, let $H_i$ be the derived graph of $D_i$. In (a copy of) $R$, replace $e_i$ with $H_i$, making the $j$th endpoint of $e_i$ and the $j$th external node of $H_i$ into the same node (for $j = 1, \ldots, |att(e_i)|$). The resulting node is external iff the $j$th endpoint was. All other nodes are kept distinct. (Again, see Figure 1a for an example with $X = X_i$ and $H = H_i$.)

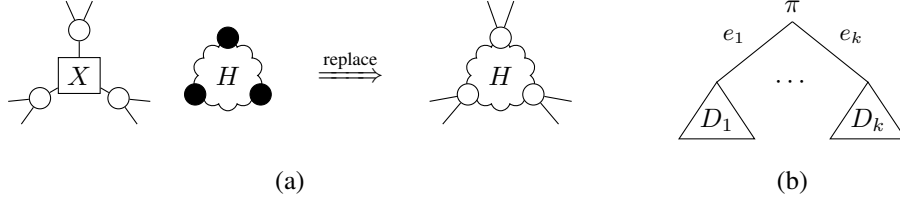

(a)                                                                    (b)

Figure 1: (a) Example of replacing a hyperedge labeled $X$ with a hypergraph fragment $H$. Here $|type(X)| = 3$, but in general, there could be any number of endpoint/external nodes, including zero. (b) A derivation tree.

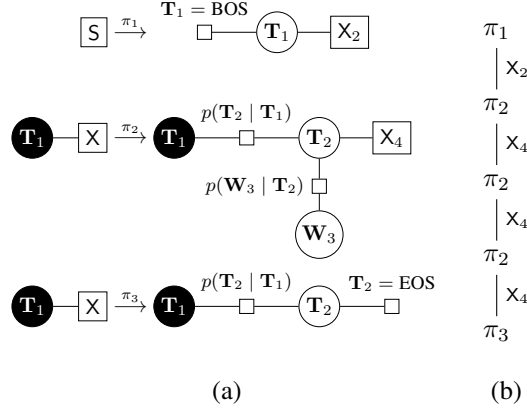

(a)                                          (b)

Figure 2: (a) A FGG generating the infinite set of unrollings of a HMM, one for each sequence length. Each rule is labeled $\pi_i$ for use in the derivation tree. (b) Derivation tree of the factor graph of Example 3. An edge from parent $\pi$ with label $X$ to child $\pi'$ means that the right-hand side of $\pi'$ replaces the edge labeled $X$ in the right-hand side of $\pi$.

From now on, when we mention a derivation $D$ in a context where a graph would be expected, the derived graph of $D$ is to be understood.

## 3 Factor Graph Grammars

**Definition 8.** A HRG for factor graphs, or a *factor graph grammar* (FGG) for short, is a HRG together with mappings $\Omega$ and $F$, as in the definition of factor graphs (Definition 2), except that $F$ is defined on terminal edge labels only.

**Example 9.** Figure 2 shows a FGG which is equivalent to a HMM. It generates an infinite number of graphs, one for each string length. Also shown is the derivation tree of the factor graph of Example 3.

Example 19 in Appendix A shows how to simulate a PCFG in Chomsky normal form as a FGG.

The graphs generated by a FGG can be viewed, together with $\Omega$ and $F$, as factor graphs, each of which defines a (not necessarily normalized) distribution over assignments. Moreover, the whole language of the FGG defines a (not necessarily normalized) distribution over derivations and assignments to the variables in them. If $D \in \mathcal{D}(G)$, then

$$w_G(D, \xi) = w_D(\xi).$$

FGGs can simulate several other formalisms for dynamically-structured models. As mentioned above (§1), they can solve two problems that previous formalisms have addressed separately.

FGGs can generate repeated substructures like plate notation (Buntine, 1994; Obermeyer et al., 2019) and dynamic graphical models (Bilmes, 2010) can. There are some structures that plate notation can describe that a FGG cannot – like the set of all restricted Boltzmann machines, which have two

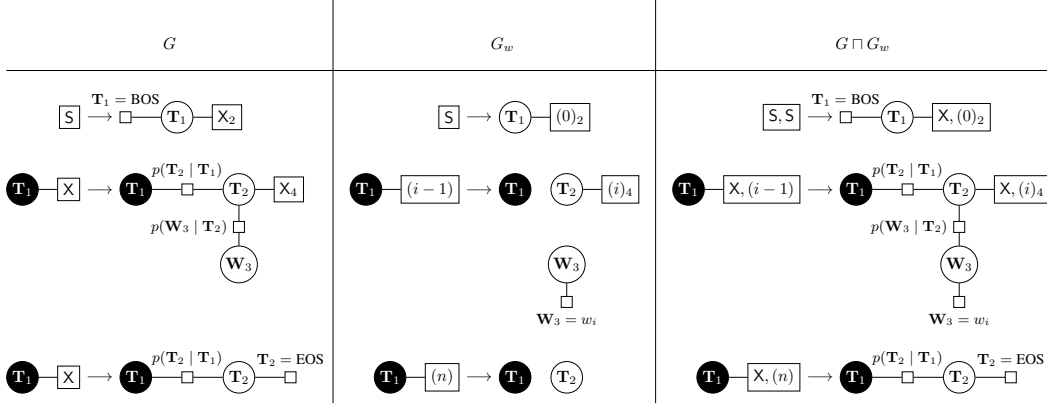

Figure 3: Illustration of the conjunction operation (Example 12). In rules with $i$ in their nonterminals, $i$ ranges from 1 to $n$ where $n = |w|$.

fully-connected layers of nodes. But these are the same structures that Obermeyer et al. (2019) try to avoid, because inference on them is (believed) intractable. FGG rules these structures out naturally.

FGGs can generate alternative substructures like case–factor diagrams (McAllester et al., 2008) and valid sum–product networks (Poon and Domingos, 2011) can; in particular, they can simulate PCFGs in such a way that inference is equivalent to the cubic-time inside and Viterbi algorithms.

**Theorem 10.** *All of the following can be converted into an equivalent FGG:*

1. *Plated factor graphs for which the sum–product algorithm of Obermeyer et al. (2019) succeeds.*

2. *Dynamic graphical models.*

3. *Case–factor diagrams.*

4. *Valid sum–product networks.*

*Proof.* See Appendix B. □

## 4   Conjunction

The preceding examples show how to use FGGs to model the probability of all tagged strings or all trees generated by a grammar. But it's common for queries to constrain some variables to fixed values, sum over some variables, and get the distribution of the remaining variables. How do such queries generalize to FGGs? For example, in a HMM, how do we compute the probability of all taggings of a given string? Or, how do we compute the marginal distribution of the second-to-last tag?

To answer such questions, we need to be able to specify a set of nodes across the graphs of a graph language, like the second-to-last tag. Our only means of doing this is to specify a particular node in a particular right-hand side, which could correspond to zero, one, or many nodes in the derived graphs. And we can modify a FGG so that a particular node in a particular right-hand side is always (say) the second-to-last tag. But we propose to factor such modifications into a separate FGG, keeping the FGG describing the model unchanged. Then the modifications can be applied using a *conjunction* operation, which we describe in this section.

Conjunction is closely related to synchronous HRGs (Jones et al., 2012), and, because HRG derivation trees are generated by regular tree grammars, to intersection/composition of finite tree automata/transducers (Comon et al., 2007). It is also similar to the PRODUCT operation on weighted logic programs (Cohen et al., 2011).

**Definition 11.** Two FGG rules are *conjoinable* if they can be written in the form

$$X_1 \to R_1 \qquad R_1 = (V, E_N \cup E_1, att_N \cup att_1, lab^V, lab_1^E, ext)$$
$$X_2 \to R_2 \qquad R_2 = (V, E_N \cup E_2, att_N \cup att_2, lab^V, lab_2^E, ext),$$

where

- $E_N$ contains only nonterminal edges, and $att_N$ is defined on $E_N$.

- $E_1$, $E_2$ contain only terminal edges, and $att_1$, $att_2$ are defined on $E_1$, $E_2$, respectively.

- $type(X_1) = type(X_2)$, and for $e \in E_N$, $type(lab_1^E(e)) = type(lab_2^E(e))$.

Then their *conjunction* is

$$\langle X_1, X_2 \rangle \to R \qquad\qquad R = (V, E_N \cup E_1 \uplus E_2, att, lab^V, lab^E, ext)$$

where $\uplus$ means that all edges in $E_1$ and $E_2$ are kept distinct while taking their union, and

$$lab^E(e) = \begin{cases} \langle lab_1^E(e), lab_2^E(e) \rangle & \text{if } e \in E_N \\ lab_1^E(e) & \text{if } e \in E_1 \\ lab_2^E(e) & \text{if } e \in E_2 \end{cases} \qquad att(e) = \begin{cases} att_N(e) & \text{if } e \in E_N \\ att_1(e) & \text{if } e \in E_1 \\ att_2(e) & \text{if } e \in E_2. \end{cases}$$

The *conjunction* of two FGGs $G_1$ and $G_2$, written as $G_1 \sqcap G_2$, is the FGG containing the conjunction of all conjoinable pairs of rules from $G_1$ and $G_2$.

**Example 12.** Our FGG for HMMs (Example 9) is repeated in Figure 3 as $G$. We can constrain the $\mathbf{W}$ variables to an observed string $w$ using another FGG, $G_w$, which has the same variables as $G$ but different factors; its nonterminal edges are the same as $G$ but with different labels. This FGG generates just one graph, whose $\mathbf{W}$ nodes spell out the string $w$. The conjunction of these two FGGs is shown in the last column ($G \sqcap G_w$). It combines the factors and nonterminal labels of $G$ and $G_w$ and generates just one graph, the HMM for string $w$.

**Example 13.** To compute the distribution of the second-to-last tag, we need a way of identifying the variable for the second-to-last tag across all graphs. We can do this by conjoining with the FGG:

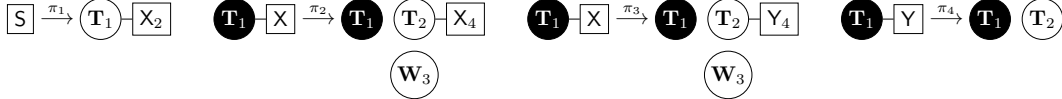

Then the second-to-last tag is always node $\mathbf{T}_1$ in the right-hand side of rule $\pi_3$. The methods of the following section can then be used to compute the distribution of this node.

Example 20 in Appendix A shows how to use conjunction to constrain a PCFG to a single input string.

## 5 Inference

Given a FGG $G$, we want to be able to efficiently compute its sum–product,

$$Z_G = \sum_{D \in \mathcal{D}(G)} \sum_{\xi \in \Xi_D} w_G(D, \xi).$$

We can answer a wide variety of queries by using the conjunction operation to constrain variables based on observations, and then computing the sum–product in various semirings: ordinary addition and multiplication would sum over assignments to the remaining variables, and the expectation semiring (Eisner, 2002) would compute expectations with respect to them. The Viterbi (max–product) semiring would find the highest-weight *derivation* and assignment, not necessarily the highest-weight *graph* and assignment, which is NP-hard (Lyngsø and Pedersen, 2002).

We consider three cases below: finite variable domains, but possibly infinite graph languages (§5.1); finite graph languages, but possibly infinite variable domains (§5.2); and infinite variable domains and graph languages (§5.3). To help characterize these cases and their subcases, we introduce the following definitions.

**Definition 14.** A FGG is *recursive* if it has an $X$-type derivation that contains an $X$-type derivation as a proper subtree; otherwise, it is *nonrecursive*. A nonrecursive FGG generates a finite set of graphs; this is a common case, because the conjunction of any FGG with a nonrecursive FGG (e.g., one describing a finite-sized observation) is nonrecursive.

A recursive FGG is *nonlinearly recursive* if it has an $X$-type derivation that contains two disjoint $X$-type derivations as proper subtrees; otherwise, it is *linearly recursive*.

A FGG is *nonreentrant* if no derivation contains two different $X$-type derivations as subtrees. Every nonreentrant FGG is nonrecursive, and any nonrecursive FGG can be made nonreentrant by duplicating rules and renaming nonterminals (though this may cause an exponential blowup in the size of the grammar).

## 5.1 Finite variable domains

When a HRG generates a graph, the derivation tree is isomorphic to a tree decomposition of the graph: each derivation tree node $\pi = (X \rightarrow R)$ corresponds to a bag of the tree decomposition containing the nodes in $R$. It follows that a HRG whose right-hand sides have at most $(k + 1)$ nodes generates graphs with treewidth at most $k$ (Bodlaender, 1998, Theorem 37). So if a FGG $G$ generates a graph $H$, computing the sum–product of $H$ by variable elimination (VE) takes time linear in the size of $H$ and exponential in $k$.

In this section, we generalize VE to compute the sum-product of all graphs generated by $G$ without enumerating them. If $G$ is nonrecursive, this is (like VE) linear in the size of $G$ and exponential in $k$; in the envisioned typical use-case, we have a fixed FGG $G$ representing the model and different FGGs $G'$ representing different observations; since conjunction cannot increase $k$, we may regard $k$ as fixed, so computing the sum–product of $G \sqcap G'$ takes time linear in the size of $G \sqcap G'$.

**Theorem 15.** *Let $G = (N, T, P, S)$ be a FGG such that for all $v$ in $G$, $|\Omega(v)| \leq m$. Let $|G|$ be the number of rules in $G$, and let $k$ be such that every right-hand side in $G$ has at most $(k + 1)$ nodes. Then $Z_G$ is the least solution of a monotone system of polynomial equations, and in particular:*

1. *If $G$ is nonrecursive, $Z_G$ can be computed in $O(|G|m^{k+1})$ time.*

2. *If $G$ is linearly recursive, $Z_G$ can be computed in $O(|G|^3 m^{3(k+1)})$ time in the worst case.*

*Proof.* The computation of the sum–product is closely analogous to the sum–product of a PCFG (Stolcke, 1995; Nederhof and Satta, 2008). We introduce some shorthand for assignments. If $\xi$ is an assignment and $v_1 \cdots v_l$ is a sequence of nodes, we write $\xi(v_1 \cdots v_l)$ for $\xi(v_1) \cdots \xi(v_l)$. If $X$ is a nonterminal and $type(X) = \ell_1 \ldots \ell_k$, we define $\Xi_X = \Omega(\ell_1) \times \cdots \times \Omega(\ell_k)$, the set of assignments to the endpoints of an edge labeled $X$.

Next, we define a system of equations whose solution gives the desired sum–product. The unknowns are $\psi_X(\xi)$ for all $X \in N$ and $\xi \in \Xi_X$, and $\tau_R(\xi)$ for all rules $(X \rightarrow R)$ and $\xi \in \Xi_X$. For all $X \in N$, let $P^X$ be the rules in $P$ with left-hand side $X$. For each $\xi \in \Xi_X$, add the equation

$$\psi_X(\xi) = \sum_{(X \rightarrow R) \in P^X} \tau_R(\xi).$$

For each right-hand side $R = (V, E_N \cup E_T, att, lab^V, lab^E, ext)$, where $E_N$ contains only nonterminal edges and $E_T$ contains only terminal edges, and for each $\xi \in \Xi_X$, add the equation

$$\tau_R(\xi) = \sum_{\substack{\xi' \in \Xi_R \\ \xi'(ext)=\xi}} \prod_{e \in E_T} F(e)(\xi'(att(e))) \prod_{e \in E_N} \psi_{lab^E(e)}(\xi'(att(e))).$$

Then $\sum_{\xi \in \Xi_X} \psi_X(\xi)$ represents the sum–product of all $X$-type derivations. In particular, the sum–product of the FGG is $\psi_S()$.

To solve these equations, construct a directed graph over nonterminals with an edge from $X$ to $Y$ iff there is a rule $X \rightarrow R$ where $R$ contains an edge labeled $Y$. For each connected component $C$ of this graph in reverse topological order:

1. If $C = \{X\}$, compute $\psi_X$ and substitute it into the other equations.

2. Else if the equations for $\psi_X$ and $\tau_R$ where $X \in C$ and $(X \rightarrow R) \in P$ are linear, solve them and substitute into the other equations (Stolcke, 1995; Goodman, 1999).

3. Else, the equations can be approximated iteratively (Goodman, 1999; Nederhof and Satta, 2008).

If $G$ is nonrecursive, the graph of nonterminals is acyclic, so case (1) always applies. The total running time is $O(|G|m^{k+1})$.

If $G$ is linearly recursive, then case (2) may also apply. In the worst case, the nonterminal graph is one connected component, corresponding to $O(|G|m^{k+1})$ unknowns. Solving the equations could involve inverting a matrix of this size, which takes $O(|G|^3 m^{3(k+1)})$ time.

If $G$ is nonlinearly recursive, any of the three cases may apply. For case (3), each iteration takes $O(|G|m^{k+1})$ time (fixed-point iteration method) or $O(|G|^3 m^{3(k+1)})$ time (Newton's method), but the number of iterations depends on $G$. $\qquad\square$

Finally, we note that we can reduce the sizes of the right-hand sides of a FGG by a process analogous to binarization of CFGs (Gildea, 2011; Chiang et al., 2013):

**Proposition 16.** *For any hypergraph fragment $R$, let $\bar{R}$ be the hypergraph formed by adding a hyperedge connecting $R$'s external nodes. Let $G$ be a HRG, $n_G$ be the total number of nodes in its right-hand sides, and $k$ be such that for every right-hand side $R$, the treewidth of $\bar{R}$ is at most $k$. Then there is an equivalent HRG with at most $n_G$ rules whose right-hand sides have at most $(k+1)$ nodes.*

*Proof.* See Appendix C. $\qquad\square$

### 5.2 Finite graph languages

Next, we show that a nonrecursive FGG can also be converted into an equivalent factor graph, such that the sum–product of the factor graph is equal to the sum–product of the FGG. This makes it possible to use standard graphical model inference techniques for reasoning about the FGG, even with infinite variable domains. However, the conversion increases treewidth in general, so when the method of Section 5.1 is applicable, it should be preferred.

The construction is similar to constructions by Smith and Eisner (2008) and Pynadath and Wellman (1998) for dependency parsers and PCFGs, respectively. Their constructions and ours encode a set of possible derivations as a graphical model, using hard constraints to ensure that every assignment to the variables corresponds to a valid derivation.

**Theorem 17.** *Let $G = (N, T, P, S)$ be a nonreentrant FGG. Let $n_G$ and $m_G$ be the total number of nodes and edges in the right-hand sides of $G$ respectively. Then $G$ can be converted into a factor graph with $O(n_G)$ variables and $O(n_G + m_G)$ factors which gives the same sum–product.*

*Proof.* We construct a factor graph that encodes all derivations of $G$. (Example 31 in Appendix D shows an example of this construction for a toy FGG.) First, we add binary variables (with label $\mathbf{B}$ where $\Omega(\mathbf{B}) = \{\text{true}, \text{false}\}$) that switch on or off parts of the factor graph (somewhat like the gates of Minka and Winn (2008)). For each nonterminal $X \in N$, we add $\mathbf{B}_X$, indicating whether $X$ is used in the derivation, and for each rule $\pi \in P$, we add $\mathbf{B}_\pi$, indicating whether $\pi$ is used.

Next, we create factors that constrain the $\mathbf{B}$ variables so that only one derivation is active at a time. We write $P^X$ for the set of rules with left-hand side $X$, and $P^{\to X}$ for the set of rules which have a right-hand side edge labeled $X$. Define the following function:

$$\text{CondOne}_l(\mathbf{B}, \mathbf{B}_1, \ldots, \mathbf{B}_l) = \begin{cases} \exists!\, i \in \{1, \ldots, l\} . \mathbf{B}_i & \text{if } \mathbf{B} = \text{true} \\ \neg(\mathbf{B}_1 \vee \cdots \vee \mathbf{B}_l) & \text{if } \mathbf{B} = \text{false} \end{cases}$$

Then we add these factors, which ensure that if one of the rules in $P^{\to X}$ is used (or $X = S$), then exactly one rule in $P^X$ is used; if no rule in $P^{\to X}$ is used (and $X \neq S$), then no rule in $P^X$ is used.

- For the start symbol $S$, add a factor $e$ with $att(e) = \mathbf{B}_S$ and $F(e)(\mathbf{B}_S) = (\mathbf{B}_S = \text{true})$.

- For $X \in N \setminus \{S\}$, let $P^{\to X} = \{\pi_1, \ldots, \pi_l\}$ and add a factor $e$ with $att(e) = \mathbf{B}_X \, \mathbf{B}_{\pi_1} \cdots \mathbf{B}_{\pi_l}$ and $F(e) = \text{CondOne}_l$.

- For $X \in N$, let $P^X = \{\pi_1, \ldots, \pi_l\}$ and add a factor $e$ with $att(e) = \mathbf{B}_X \, \mathbf{B}_{\pi_1} \cdots \mathbf{B}_{\pi_l}$ and $F(e) = \text{CondOne}_l$.

Next, define the function:

$$\text{Cond}(\mathbf{B}, x) = \begin{cases} x & \text{if } \mathbf{B} = \text{true} \\ 1 & \text{otherwise.} \end{cases}$$

For each rule $\pi \in P$, where $\pi = (X \to R)$ and $R = (V, E_N \cup E_T, att, lab^V, lab^E)$, we construct a "cluster" $C_\pi$ of variables and factors:

- For each $v \in V$, add a variable $v'$ with the same label to $C_\pi$. Also, add a factor with endpoints $\mathbf{B}_\pi$ and $v'$ and function $\text{CondNormalize}_{v'}(\mathbf{B}_\pi, v')$, defined to equal $\text{Cond}(\neg\mathbf{B}_\pi, p(v'))$, where $p$ is any probability distribution over $\Omega(v')$. This ensures that if $\pi$ is not used, then $v'$ will sum out of the sum–product neatly.
- For each $e \in E_T$ where $att(e) = v_1 \cdots v_k$, add a new edge $e'$ with $att(e') = \mathbf{B}_\pi v_1' \cdots v_k'$ and function $\text{CondFactor}_{e'}(\mathbf{B}_\pi, v_1', \ldots, v_k')$, defined to equal $\text{Cond}(\mathbf{B}_\pi, F(e)(v_1', \ldots, v_k'))$.

Next, for each $X \in N$, let $l = |type(X)|$. We create a cluster $C_X$ containing variables $v_{X,i}$ for $i = 1, \ldots, l$, which represent the endpoints of $X$, such that $lab^V(v_{X,i}) = type(X)_i$. We give each an accompanying factor with endpoints $\mathbf{B}_\pi$ and $v_{X,i}$ and function $\text{CondNormalize}_{v_{X,i}}$.

These clusters are used by the factors below, which ensure that if two variables are identified during rewriting, they have the same value. Define $\text{CondEquals}(\mathbf{B}, v, v') = \text{Cond}(\mathbf{B}, v = v')$.

- For each $\pi \in P^{\to X}$, let $v_1, \ldots, v_l$ be the endpoints of the edge in $\pi$ labeled $X$. (By non-reentrancy, there can be only one such edge.) For $i = 1, \ldots, l$, create a factor $e$ where $att(e) = \mathbf{B}_\pi v_{X,i} v_i$ and $F(e) = \text{CondEquals}$.
- For each $\pi \in P^X$, let $ext$ be the external nodes of $\pi$. For $i = 1, \ldots, l$, create a factor $e$ where $att(e) = \mathbf{B}_\pi v_{X,i} ext_i$ and $F(e) = \text{CondEquals}$.

The resulting graph has $|N| + |P|$ binary variables, $n_G$ variables in the clusters $C_\pi$, and $\sum_{X \in N} |type(X)| \le n_G$ variables in the clusters $C_X$, so the total number of variables is in $O(n_G)$. It has $m_G$ $\text{CondFactor}_e$ factors, $n_G + \sum_{X \in N} |type(X)| \le 2n_G$ $\text{CondNormalize}_v$ factors, $2|N|$ $\text{CondOne}_l$ factors, and $2\sum_{(X \to R) \in P} |ext_R| \le 2n_G$ $\text{CondEquals}$ factors, so the total number of factors is in $O(n_G + m_G)$.

Appendix D contains more information on this construction, including an example, a detailed proof that the sum–product is preserved, and a discussion of inference on the resulting graph. □

## 5.3 Infinite variable domains, infinite graph languages

Finally, if we allow both (countably) infinite domains and infinite graph languages, then computing the sum–product is undecidable. This has already been observed even for single factor graphs with infinite variable domains (Dreyer and Eisner, 2009), but we show further that this can be done using a minimal inventory of factors.

**Theorem 18.** *Let $G$ be a FGG whose variable domains are $\mathbb{N}$ and whose factors only use the successor relation and equality with zero. It is undecidable whether the sum–product of $G$ is zero.*

*Proof.* By reduction from the halting problem for Turing machines. See Appendix E. □

## 6 Conclusion

Factor graph grammars are a powerful way of defining probabilistic models that permits practical inference. We plan to implement the algorithms described in this paper as differentiable operations and release them as open-source software. We will also explore techniques for optimizing inference in FGGs, for example, by automatically modifying rules to reduce their treewidth (Bilmes, 2010) or reducing the cost of matrix inversions in Theorem 15 (Nederhof and Satta, 2008). Another important direction for future work is the development of approximate inference algorithms for FGGs.

## Broader Impact

This research is of potential benefit to anyone working with structured probability models, including latent-variable neural networks. As this research is purely theoretical, we are not aware of any direct negative impacts.

## Acknowledgments and Disclosure of Funding

We would like to thank the anonymous reviewers, especially Reviewer 3, for making numerous suggestions for improvement. We also thank Antonis Anastasopoulos, Justin DeBenedetto, Wes Filardo, Chung-Chieh Shan, and Xing Jie Zhong for their feedback.

This material is based upon work supported by the National Science Foundation under Grant No. 2019291. Any opinions, findings, and conclusions or recommendations expressed in this material are those of the authors and do not necessarily reflect the views of the National Science Foundation.

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
