[Supplementary Material]

## A  Simulating PCFGs

**Example 19.** Below is a FGG for derivations of a PCFG in Chomsky normal form. The start symbol of the FGG is $S'$ and the start symbol of the PCFG is $S$. Random variables $\mathbf{N}$ range over nonterminal symbols of the PCFG, and random variables $\mathbf{W}$ range over terminal symbols.

**Example 20.** We can conjoin the FGG of Example 19 with the following FGG to constrain it to an input string $w$, with $n = |w|$, $0 \leq i < j < k \leq n$, and $1 \leq l \leq n$:

The resulting rules have a total of $O(n^3)$ variables in their right-hand sides. The largest right-hand side has 3 variables, so $k = 2$. The variables range over nonterminals, so $m = |N|$ where $N$ is the CFG's nonterminal alphabet. Therefore, running the algorithm of Theorem 15 on this FGG takes $O(n_G m^{k+1}) = O(|N|^3 n^3)$ time, which is the same as the CKY algorithm. This construction generalizes easily to CFGs not in Chomsky normal form; applying Lemma 16 would keep the inference complexity down to $O(n^3)$ (or $O(n^2)$ for a linear CFG).

## B  Relationship to other formalisms

### B.1  Plate diagrams

Plate diagrams are extensions of graphs that describe repeated structure in Bayesian networks (Buntine, 1994) or factor graphs (Obermeyer et al., 2019). A plate is a subset of variables/factors, together with a count $M$, indicating that the variables/factors inside the plate are to be replicated $M$ times. But there cannot be edges between different instances of a plate.

**Definition 21.** A *plated factor graph* or *PFG* (Obermeyer et al., 2019) is a factor graph $H = (V, E)$ together with a finite set $B$ of *plates* and a function $P : V \cup E \to 2^B$ that assigns each variable and factor to a set of plates. If $b \in P(v)$ and $e$ is incident to $v$, then $b \in P(e)$.

The *unrolling* of $H$ by $M : B \to \mathbb{N}$ is the factor graph that results from making $M(b)$ copies of every node $v$ such that $b \in P(v)$ and every edge $e$ such that $e \in P(e)$.

Obermeyer et al. (2019) give an algorithm for computing the sum–product of a PFG. It only succeeds on some PFGs. An example for which it fails is the set of all restricted Boltzmann machines (fully-connected bipartite graphs); one of their main results is to characterize the PFGs for which their algorithm succeeds. Below, we show how to convert these PFGs to FGGs.

**Proposition 22.** *Let $H$ be a PFG. If the sum–product algorithm of Obermeyer et al. (2019) succeeds on $H$, then there is a FGG $G$ such that for any $M : B \to \mathbb{N}$, there is a FGG $G_M$ such that $G \sqcap G_M$ generates one graph, namely the unrolling of $H$ by $M$.*

*Proof.* We just describe how to construct $G \sqcap G_M$ directly; hopefully, it should be clear how to construct $G$ and $G_M$ separately ($G$ has factors but not counts; $G_M$ has counts but not factors). Algorithm 1 converts $H$ and $M$ to $G \sqcap G_M$. It has the same structure as the sum–product algorithm of Obermeyer et al. (2019) and therefore works on the same class of PFGs. □

**Algorithm 1** Procedure for converting a PFG $H$ and count assignment $M$ to a FGG.

---

**while** $E \neq \emptyset$ **do**
  let $e = \arg\max_{e \in E} |P(e)|$ (breaking ties arbitrarily) and $L = P(e)$
  let $H_L$ be the subgraph of nodes and edges of $H$ whose plate set is $L$
  **for** each connected component $H_c$ of $H_L$ **do**
    let $V_f$ be the variables not in $H_c$ but incident to factors in $H_c$
    let $L' = \cup_{v \in V_f} P(v)$
    **if** $L = L'$ **then**
      **error**
    let $X$ be a fresh nonterminal
    let $n = \prod_{b \in L \setminus L'} M(b)$
    replace $H_c$ with an edge with label $X^n$ and endpoints $V_f$
    **for** $i \leftarrow n, \dots, 1$ **do**
      create rule $X^i \rightarrow R$ where $R$ has:
        • internal nodes and edges from $H_c$
        • external nodes $V_f$
        • an edge with label $X^{i-1}$ and endpoints $V_f$
    create rule $X^0 \rightarrow R$ where $R$ has external nodes $V_f$ and no other nodes/edges
  create rule $S \rightarrow H$

---

If the algorithm of Obermeyer et al. (2019) fails on a PFG, there might not be an equivalent FGG. In particular, FGGs cannot generate the set of RBMs, because a $m \times n$ RBM has treewidth $\min(m, n)$, so the set of all RBMs has unbounded treewidth and can't be generated by a HRG.

Although, in this respect, FGGs are less powerful than PFGs, we view this as a strength, not a weakness. Because FGGs inherently generate graphs of bounded treewidth, our sum–product algorithm (Theorem 15) works on all FGGs, and no additional constraints are needed to guarantee efficient inference.

**Example 23.** The following PFG is from Obermeyer et al. (2019):

Converting to a FGG produces the following rules (in order of their construction by the above algorithm):

## B.2 Dynamic graphical models

For simplicity, we only consider binary factors, which we draw as directed edges, and we ignore edge labels.

**Definition 24.** A *dynamic graphical model* or *DGM* (Bilmes, 2010) is a tuple $(H_1, H_2, H_3, E_{12}, E_{22}, E_{23})$, where the $H_i = (V_i, E_i)$ are factor graphs and the $E_{ij} \subseteq V_i \times V_j$ are sets of edges from $H_i$ to $H_j$.

A DGM specifies how to construct, for any length $n \geq 2$, a factor graph

$$H^n = (V_1 \cup V_2 \times \{1, \ldots, n\} \cup V_3, E),$$

where $E$ is defined by:

- If $(u, v) \in E_{12}$, add an edge from $u$ to $(v, 1)$.
- If $(u, v) \in E_{22}$, add an edge from $(u, i - 1)$ to $(v, i)$ for all $1 < i \leq n$.
- If $(u, v) \in E_{23}$, add an edge from $(u, n)$ to $v$.

**Proposition 25.** *Given a DGM $D = (H_1, H_2, H_3, E_{12}, E_{22}, E_{23})$, there is a FGG $G$ such that for any count $n \geq 2$, there is another FGG $G_n$ such that $G \sqcap G_n$ generates exactly one graph, the unrolling of $D$ by $n$.*

*Proof.* Again, we give an algorithm for constructing $G \sqcap G_n$, and hopefully, it should be clear how to construct $G$ and $G_n$ separately. Create the following rules:

- $\mathsf{S} \to R$, where $R$ contains
    - Nodes and edges from $H_1$, $H_2$, and $E_{12}$
    - An edge labeled $\mathsf{A}^{n-1}$ and endpoints $\{u \mid (u, v) \in E_{22}\}$.
- $\mathsf{A}^i \to R$, where $R$ contains
    - Nodes and edges from $H_2$
    - If $(u, v) \in E_{22}$, $R$ has an external node $u'$
    - For each $(u, v) \in E_{22}$, an edge from $u'$ to $v$
    - An edge labeled $\mathsf{A}^{i-1}$ and endpoints $\{u \mid (u, v) \in E_{22}\}$.
- $\mathsf{A}^1 \to R$, where $R$ contains
    - Nodes and edges from $H_2$, $H_3$, and $E_{23}$
    - If $(u, v) \in E_{22}$, $R$ has an external node $u'$
    - For each $(u, v) \in E_{22}$, an edge from $u'$ to $v$.

$\square$

**Example 26.** Bilmes (2010) give the following example of a DGM. All factors have two endpoints, and we draw them as directed edges instead of the usual squares. We draw the edges in $E_{22}$ with dotted lines.

The resulting FGG:

where, in the middle rule, $1 < i < n$.

Running the algorithm of Theorem 15 would *not* be guaranteed to achieve the same time complexity as that of (Bilmes and Bartels, 2003), which searches through alternative ways of dividing the unrolled factor graph into time slices.

## B.3 Case–factor diagrams and sum–product networks

Case–factor diagrams (McAllester et al., 2008) and sum–product networks (Poon and Domingos, 2011) are compact representations of probability distributions over assignments to Boolean variables. They generalize both Markov networks and PCFGs.

Both formalisms represent models as rooted directed acyclic graphs (DAGs), with edges directed away from the root, in which some nodes mention variables. If $D$ is a DAG, for any node $v \in D$, let $scope(v)$ be the set of variables mentioned in $v$ or any descendant of $v$.

**Definition 27.** A *case–factor diagram (CFD) model* is a pair $(D, \Psi)$, where $D$ is a rooted DAG with root $r$, each of whose nodes is one of the following:

- case$(x)$ with two children $v_1$ and $v_2$, where $x$ is a variable not in $scope(v_1) \cup scope(v_2)$.
- factor with two children $v_1$ and $v_2$, where $scope(v_1) \cap scope(v_2) = \emptyset$.
- unit with no children.
- empty with no children.

And $\Psi : scope(r) \to \mathbb{R}_{\geq 0}$ assigns a cost to each variable in $scope(r)$.

A CFD model defines a probability distribution over assignments to its variables. We compute quantities $q(v, \xi)$ and $Z(v)$ for each node $v$ as follows. Let $v_1, v_2$ be the children of $v$, if any.

$$
\begin{aligned}
&v = \mathsf{case}(x) && q(v, \xi) = \begin{cases} e^{-\Psi(x)} \, q(v_1, \xi) & \text{if } \xi(x) = 1 \\ q(v_2, \xi) & \text{if } \xi(x) = 0 \end{cases} && Z(v) = e^{-\Psi(x)} \, Z(v_1) + Z(v_2) \\
&v = \mathsf{factor} && q(v, \xi) = q(v_1, \xi) \, q(v_2, \xi) && Z(v) = Z(v_1) \, Z(v_2) \\
&v = \mathsf{unit} && q(v, \xi) = 1 && Z(v) = 1 \\
&v = \mathsf{empty} && q(v, \xi) = 0 && Z(v) = 0
\end{aligned}
$$

Define $q(\xi) = q(r, \xi)$ and $Z = Z(r)$. Then $P(\xi) = q(\xi)/Z$.

**Proposition 28.** *If $(D, \Psi)$ is a CFD model, there is a FGG $G$ such that $(D, \Psi)$ and $G$ have the same sum–product, and for any assignment $\xi$ of $(D, \Psi)$, there is a FGG $G_\xi$ such that the sum–product of $G \wedge G_\xi$ equals $q(\xi)$.*

*Proof.* Given a CFD, we can construct a FGG where each node $v$ of the CFD becomes a different nonterminal symbol $\mathsf{D}_v$:

| node | $G$ | $G_\xi$ |
|---|---|---|

$v = \mathsf{case}(x)$

$$\boxed{\mathsf{D}_v} \longrightarrow \;\; \textcircled{x} \quad \underset{e^{-\Psi(x)}}{\square} \;\; \boxed{\mathsf{D}_{v_1}} \qquad\qquad \boxed{\mathsf{D}_v} \longrightarrow \;\; \textcircled{x} \quad \boxed{\mathsf{D}_{v_1}}$$
$$\underset{x=1}{\square} \qquad\qquad\qquad\qquad\qquad \underset{x=\xi(x)}{\square}$$

$$\boxed{\mathsf{D}_v} \longrightarrow \;\; \textcircled{x} \qquad \boxed{\mathsf{D}_{v_2}} \qquad\qquad \boxed{\mathsf{D}_v} \longrightarrow \;\; \textcircled{x} \quad \boxed{\mathsf{D}_{v_2}}$$
$$\underset{x=0}{\square} \qquad\qquad\qquad\qquad\qquad \underset{x=\xi(x)}{\square}$$

$v = \mathsf{factor}$
$$\boxed{\mathsf{D}} \longrightarrow \boxed{\mathsf{D}_{v_1}} \; \boxed{\mathsf{D}_{v_2}} \qquad\qquad \boxed{D} \longrightarrow \boxed{\mathsf{D}_{v_1}} \; \boxed{\mathsf{D}_{v_2}}$$

$v = \mathsf{unit}$
$$\boxed{\mathsf{unit}} \longrightarrow \emptyset \qquad\qquad\qquad \boxed{\mathsf{unit}} \longrightarrow \emptyset$$

We do not create any rule with left-hand side empty, so that any derivations that generate empty fail. $\qquad\square$

The number of rules in $G$ is the number of nodes in $D$. Computing its sum–product is linear in the number of rules, just as computing the sum–product of $D$ is linear in the number of nodes.

**Definition 29.** A valid *sum–product network* (SPN) is a rooted DAG whose nodes are each either:

- $\mathsf{sum}(\lambda_1, \lambda_2)$ with two children $v_1$ and $v_2$, where $scope(v_1) = scope(v_2)$.
- $\mathsf{product}$ with two children $v_1$ and $v_2$ such that no variable appears in one and negated in the other.
- $x$ or $\bar{x}$ with no children.

A valid SPN defines a distribution over assignments to its variables. For each node $v$, let $v_1, v_2$ be the children of $v$, if any.

$$
\begin{aligned}
v &= \mathsf{sum}(\lambda_1, \lambda_2) & q(v, \xi) &= \lambda_1 q(v_1, \xi) + \lambda_2 q(v_2, \xi) \\
v &= \mathsf{product} & q(v, \xi) &= q(v_1, \xi)\, q(v_2, \xi) \\
v &= x & q(v, \xi) &= \xi(x) \\
v &= \bar{x} & q(v, \xi) &= 1 - \xi(x)
\end{aligned}
$$

Converting a valid SPN to a FGG is straightforward, but the resulting FGG has a separate node for each occurrence of a variable $x$. The syntactic constraints in the definition of valid SPN ensure that in any graph with nonzero weight, all occurrences of $x$ have the same value.

**Proposition 30.** *Any valid SPN $S$ can be converted into a FGG $G$ such that $S$ and $G$ have the same sum–product, and for any assignment $\xi$ of $S$, there is a FGG $G_\xi$ such that the sum–product of $G \wedge G_\xi$ equals $q(\xi)$.*

*Proof.* We construct a FGG where each node $v$ becomes a different nonterminal symbol $D_v$:

| node | $G$ | $G_\xi$ |
|------|-----|---------|
| $v = x$ | $\boxed{D_v} \longrightarrow (x)\!\!-\!\!\square$ $\quad x = 1$ | $\boxed{D_v} \longrightarrow (x)\!\!-\!\!\square$ $\quad x = \xi(x)$ |
| $v = \bar{x}$ | $\boxed{D_v} \longrightarrow (x)\!\!-\!\!\square$ $\quad x = 0$ | $\boxed{D_v} \longrightarrow (x)\!\!-\!\!\square$ $\quad x = \xi(x)$ |
| $v = \mathsf{sum}(\lambda_1, \lambda_2)$ | $\boxed{D_v} \longrightarrow \underset{\lambda_1}{\square}\ \boxed{D_{v_1}}$ <br> $\boxed{D_v} \longrightarrow \underset{\lambda_2}{\square}\ \boxed{D_{v_2}}$ | $\boxed{D_v} \longrightarrow \boxed{D_{v_1}}$ <br> $\boxed{D_v} \longrightarrow \boxed{D_{v_2}}$ |
| $v = \mathsf{product}$ | $\boxed{D_v} \longrightarrow \boxed{D_{v_1}}\ \boxed{D_{v_2}}$ | $\boxed{D_v} \longrightarrow \boxed{D_{v_1}}\ \boxed{D_{v_2}}$ |

□

The number of rules in $G$ is the number of nodes in $S$. Computing its sum–product is linear in the number of rules, just as computing the sum–product of $S$ is linear in the number of nodes.

Further variations of SPNs have been proposed, in particular to generate repeated substructures (Stuhlmüller and Goodman, 2012; Melibari et al., 2016). Factored SPNs (Stuhlmüller and Goodman, 2012) are especially closely related to FGGs, in that they allow one part of a SPN to "reference" another, which is analogous to a nonterminal-labeled edge in a FGG.

CFDs and SPNs present a rather different, lower-level view of a model than the other formalisms surveyed here do. Whereas factor graphs and the other formalisms represent the model's *variables* and the *dependencies* among them, CFDs and SPNs (including factored SPNs) represent the *computation* of the sum-product. For instance, converting a factor graph $H$ to a CFD or SPN requires forming a tree decomposition of $H$ (McAllester et al., 2008), and the resulting CFD/SPN's structure is that of the tree decomposition, not of $H$.

FGGs, in a sense, combine both points of view. Their derived graphs represent a model's variables and dependencies, while their derivation trees represent the computation of the sum-product. Thus, a factor graph $H$ can be trivially converted into a FGG $S \to H$, and, as can be seen in the translations given above, a CFD or SPN can also be converted to a FGG while preserving its structure.

## C  Proof of Proposition 16

Let $H = (V, E)$ be a hypergraph. Recall that a *tree decomposition* of $H$ is a tree whose nodes are called *bags*, to each of which is associated a set of nodes, $V_B \subseteq V$, and (nonstandardly) a set of edges, $E_B \subseteq E$. The bags must satisfy the properties:

- Node cover: $\bigcup_B V_B = V$.

- Edge cover: for every edge $e \in E$, there is exactly one bag $B$ such that $e \in E_B$ and $att(e) \subseteq V_B$.

- Running intersection: if $v \in V_{B_1}$ and $v \in V_{B_2}$, then for every bag $B$ between $B_1$ and $B_2$, $v \in V_B$.

The *width* of a tree decomposition is $\max_B |V_B| - 1$, and the *treewidth* of $H$ is the minimum width of any tree decomposition of $H$. A tree decomposition can always be made to have at most $n$ nodes without changing its width (Bodlaender, 1993).

Chiang et al. (2013) give a parsing algorithm for HRGs that matches right-hand sides incrementally using their tree decompositions. They observe that this is related to the concept of binarization of context-free grammars. Here, we make this connection explicit by showing how to factorize a HRG.

For every rule $(X \to R)$, where $\bar{R}$ has $n_R$ nodes and treewidth at most $k$, form a tree decomposition of $\bar{R}$ with $n_R - k \le n_R$ bags. Let the root of the tree decomposition be the bag containing all the external nodes of $R$. For each bag $B$, construct a rule $X_B \to R_B$ as follows.

- If $B$ is the root bag, $X_B = X$; otherwise, $X_B$ is a fresh nonterminal symbol.

- Add all nodes in $V_B$ and edges in $E_B$ to $R_B$.

- If $B$ is the root bag, $R_B$'s external nodes are the same as $R$'s; if $B$ has parent $P$, let $R_B$'s external nodes be $V_P \cap V_B$.

- For each child bag $B_i$, add a hyperedge with label $X_{B_i}$ and endpoints $V_B \cap V_{B_i}$.

This new FGG generates the same language as $G$. The number of rules is at most $\sum_{(X \to R) \in G} n_R = n_G$. Every right-hand side has at most $(k+1)$ nodes.

# D  Supplement to Theorem 17

## D.1  An example

**Example 31.** We show how to construct the factor graph corresponding to the following simple, nonreentrant FGG:

This grammar generates just two graphs:

Applying the construction from Theorem 17 gives the factor graph shown in Figure 4.

$\square$ $\mathbf{B_S} = \text{true}$

S

$\mathbf{B_S}$

$\square$ $\text{CondOne}(\mathbf{B_S}, \mathbf{B}_{\pi_1}, \mathbf{B}_{\pi_2})$

$\mathbf{A}_1$ $\mathbf{B}_2$

$\mathbf{A}_4$

$\pi_1$

$\mathbf{B}_{\pi_1}$

$\mathbf{B}_{\pi_2}$

$\mathbf{A}_1$ $\mathbf{B}_2$

$\pi_2$

$\square$ $\text{CondOne}(\mathbf{B_X}, \mathbf{B}_{\pi_1})$

$\mathbf{A}_1$ $\mathbf{B}_2$

$\mathbf{A}_4$

X

$\mathbf{B_X}$

$\square$ $\text{CondOne}(\mathbf{B_X}, \mathbf{B}_{\pi_3})$

$\mathbf{A}_1$ $\mathbf{B}_2$

$f$

$\mathbf{A}_4$

$\pi_3$

$\mathbf{B}_{\pi_3}$

$\square$ $\text{CondOne}(\mathbf{B_Y}, \mathbf{B}_{\pi_2}, \mathbf{B}_{\pi_3})$

$\mathbf{A}$ $\mathbf{B}$

$\mathbf{B_Y}$

Y

$\square$ $\text{CondOne}(\mathbf{B_Y}, \mathbf{B}_{\pi_4})$

$g$

$\mathbf{A}$ $\square$ $\mathbf{B}$

$\mathbf{B}_{\pi_4}$

$\pi_4$

Figure 4: The FGG of Example 31, converted to a single factor graph using the construction of Theorem 17. Some detail has been omitted to reduce clutter. The edges from clusters to $\mathbf{B}$ variables are "meta-tentacles" that stand for a tentacle from every factor inside the cluster to the $\mathbf{B}$ variable. We draw CondNormalize and CondEquals factors as smaller squares and omit their names. Lastly, rather than writing out "CondFactor", we use the name of the original factor function ($f$ or $g$).

## D.2 Complexity of inference

As noted in Section 5.2, the purpose of this conversion to a single factor graph is to make inference possible with infinite variable domains; after converting to a factor graph, existing, possibly approximate, inference methods can be applied. But with finite variable domains, an algorithm like variable elimination would not be appropriate because this conversion has the potential to increase treewidth dramatically.

In the proof of Theorem 15, we constructed the *nonterminal graph*, which has a node for every nonterminal and an edge from $X$ to $Y$ iff there is a rule $X \to R$ where $R$ has an edge labeled $Y$. For a nonreentrant FGG, the nonterminal graph is always a DAG. If, for each $X \in N \setminus S$, $X$ appears in the right-hand side of exactly one rule, then the nonterminal graph is a tree.

When the nonterminal graph is a tree, we can construct a tree decomposition by making one bag for each cluster, and one bag for each CondOne factor. The bag for a cluster $C$ contains all the variables in $C$, along with $\mathbf{B}_C$ and all the CondNormalize and CondFactor edges associated with $C$. The bag for a CondOne factor will contain all the $\mathbf{B}$ variables used by that CondOne factor, all the CondEquals edges connecting clusters involved in that CondOne factor, and all the variables connected to those CondEquals edges.

Exact inference on this tree decomposition is very similar to the algorithm described in Theorem 15. However, a naïve application of variable elimination will still be less efficient than that algorithm, since the CondOne factors connect $|P^X| + 1$ binary variables, requiring a loop over $2^{|P^X|+1}$ assignments. All but $|P^X| + 1$ of these assignments have zero weight, so in fact we can process these factors much faster; modifying the variable elimination algorithm to account for this and the CondEquals constraints would give us something almost identical to the algorithm of Theorem 15.

In the DAG case, this simple tree decomposition is not possible. The factor graph $H$ has the nonterminal graph as a minor, so the treewidth of the nonterminal graph is a lower bound on the treewidth of $H$ (Bodlaender, 1998, Lemma 16). In the worst case, this could be $|N|$.

## D.3 Detailed proof of correctness

If $G$ is a FGG and $H$ is the factor graph that results from the construction of Theorem 17, we can show that they have the same sum–product $Z_G = Z_H$.

The sum–product $Z_H$ can be computed in the usual way, by summing over all assignments to the variables and, for each assignment, taking the product over all of the factors:

$$Z_H = \sum_{\xi \in \Xi_H} \prod_{e \in H} F(e)(\xi(e)).$$

The summation over assignments $\xi$ includes many possible settings of the $\mathbf{B}$ variables. But the CondOne factors tell us that, if the assignment to the $\mathbf{B}$ variables does not give us a valid derivation, then the weight of that assignment will be 0. Therefore, we only need to sum over assignments to the $\mathbf{B}$ variables which represent a valid derivation, and so we can express the sum–product using a sum over derivations rather than a sum over assignments to $\mathbf{B}$ variables. Let $\xi_{\mathbf{B}}$ represent the assignment to the $\mathbf{B}$ variables. Then:

$$Z_H = \sum_{D \in \mathcal{D}(G)} \sum_{\substack{\xi \in \Xi_H \\ \xi_{\mathbf{B}} \text{ consistent with } D}} \prod_{e \in H} F(e)(\xi(e)).$$

(Note that the product over $e \in H$ can ignore all CondOne factors, since when the assignment to the $\mathbf{B}$ variables is consistent with some derivation, they all have value 1.)

We can associate a derivation $D$ with the subset of clusters in $H$ corresponding to the nonterminals and rules which were used in the derivation; call this $\mathcal{C}_D$. For any $D$, all the variables in $H$ are divided into three parts: those that belong to clusters in $\mathcal{C}_D$ (call this $V_D$), those that belong to clusters not in $\mathcal{C}_D$ (call this $V_{\overline{D}}$), and the $\mathbf{B}$ variables (which don't belong to any cluster). Let $\xi_{\mathbf{B},D}$ be the unique assignment to the $\mathbf{B}$ variables that is consistent with $D$. Let $\Xi_D$ be the set of all assignments extending $\xi_{\mathbf{B},D}$ with assignments to $V_D$, and let $\Xi_{\overline{D}}$ be the set of all assignments extending $\xi_{\mathbf{B},D}$ with assignments to $V_{\overline{D}}$.

Let $E_D$ be the set of factors involving a variable in $V_D$, and let $E_{\overline{D}}$ be the set of factors involving a variable in $V_{\overline{D}}$. Because any factors between $V_D$ and $V_{\overline{D}}$ are CondEquals factors with value 1 (since their $\mathbf{B}$ variable is false), we can ignore them. Similarly, the only factors which don't involve either $V_D$ or $V_{\overline{D}}$ are the CondOne factors, which we are already ignoring. This allows us to rewrite the sum–product as

$$Z_H = \sum_{D \in \mathcal{D}} \underbrace{\left( \sum_{\xi \in \Xi_D} \prod_{e \in E_D} F(e)(\xi(e)) \right)}_{Z_D} \underbrace{\left( \sum_{\xi \in \Xi_{\overline{D}}} \prod_{e \in E_{\overline{D}}} F(e)(\xi(e)) \right)}_{Z_{\overline{D}}}.$$

Consider $Z_{\overline{D}}$ first. All CondFactor and CondEquals factors in $E_{\overline{D}}$ have value 1 and can be ignored, leaving only CondNormalize factors. Because these place a probability distribution $p_v$ on each variable $v$ in an unused cluster, those variables all sum out:

$$
\begin{aligned}
Z_{\overline{D}} &= \sum_{\xi \in \Xi_{\overline{D}}} \prod_{C_X \not\in \mathcal{C}_D} \prod_{v \in C_X} \text{CondNormalize}_v(\mathbf{B}_X, v) \prod_{C_\pi \not\in \mathcal{C}_D} \prod_{v \in C_\pi} \text{CondNormalize}_v(\mathbf{B}_\pi, v) \\
&= \sum_{\xi \in \Xi_{\overline{D}}} \prod_{C_X \not\in \mathcal{C}_D} \prod_{v \in C_X} p_v(\xi(v)) \prod_{C_\pi \not\in \mathcal{C}_D} \prod_{v \in C_\pi} p_v(\xi(v)) \\
&= \prod_{C_X \not\in \mathcal{C}_D} \prod_{v \in C_X} \left( \sum_{x \in \Omega(v)} p_v(x) \right) \prod_{C_\pi \not\in \mathcal{C}_D} \prod_{v \in C_\pi} \left( \sum_{x \in \Omega(v)} p_v(x) \right) \\
&= 1.
\end{aligned}
$$

Now consider $Z_D$. All CondNormalize factors in $E_D$ have value 1 and can be ignored, leaving only CondEquals and CondFactor factors. Let $H_D$ be the derived graph of $D$. We can think of the derivation as merging pairs of nodes in $V_D$, so that a single node $v \in H_D$ may correspond to several "copies" in $V_D$. However, the CondEquals constraints ensure that all copies of $v$ have the same value. Therefore, instead of summing over the assignments to $V_D$, we can simply sum over the assignments to $H_D$ (and omit CondEquals factors):

$$
\begin{aligned}
Z_D &= \sum_{\xi \in \Xi_{H_D}} \prod_{C_\pi \in \mathcal{C}_D} \prod_{e \in \pi} \text{CondFactor}_e(\mathbf{B}_\pi, \xi(att(e))) \\
&= \sum_{\xi \in \Xi_{H_D}} \prod_{C_\pi \in \mathcal{C}_D} \prod_{e \in \pi} F(e)(\xi(att(e))) \\
&= \sum_{\xi \in \Xi_{H_D}} \prod_{e \in H_D} F(e)(\xi(att(e))).
\end{aligned}
$$

So, finally, the sum–product of $H$ can be rewritten as:

$$
\begin{aligned}
Z_H &= \sum_{D \in \mathcal{D}(G)} \sum_{\xi \in \Xi_{H_D}} \prod_{e \in H_D} F(e)(\xi(att(e))) \\
&= \sum_{D \in \mathcal{D}(G)} \sum_{\xi \in \Xi_{H_D}} w_G(D, \xi) \\
&= Z_G.
\end{aligned}
$$

## E   Proof of Theorem 18

Let $\Gamma$ be a finite alphabet containing a blank symbol (␣), and let $k = |\Gamma|$. Number the symbols in $\Gamma$ as $\gamma_0 = $ ␣, $\gamma_1, \gamma_2, \ldots, \gamma_{k-1}$. Define an encoding for strings over $\Gamma$:

$$
\begin{aligned}
\langle \epsilon \rangle &= 0 \\
\langle \gamma_i w \rangle &= i + k \cdot \langle w \rangle.
\end{aligned}
$$

Note that strings that differ only in the number of trailing blanks have the same encoding.

We write $x \mathbin{/\!/} k$ for $\lfloor x/k \rfloor$ and $x \mathbin{\%} k = x - x \mathbin{/\!/} k \cdot k$.

Let $M$ be a Turing machine with doubly-infinite tape, input alphabet $\Sigma$, tape alphabet $\Gamma$, start state $q_0$, transition function $\delta$, accept state $q_{\text{accept}}$, and reject state $q_{\text{reject}}$. For any input string $w \in \Sigma^*$, construct the following rules, where the **q** nodes track the Turing machine's state, the **u** nodes track the reverse of the tape to the left of the head, and the **v** nodes track the tape from the head rightward:

$$\mathbf{u}_1 = 0 \qquad \mathbf{q}_2 = q_0 \qquad \mathbf{v}_3 = \langle w \rangle$$

$$\boxed{\mathrm{S}} \longrightarrow$$

$$\mathbf{q}_2 \in \{q_{\text{accept}}, q_{\text{reject}}\}$$

For each transition $\delta(q, a) = (r, b, \mathrm{L})$:

$$\longrightarrow$$

$$\mathbf{u}_5 = \mathbf{u}_1 \mathbin{/\!/} k \qquad \mathbf{q}_2 = q \qquad \mathbf{v}_3 \mathbin{\%} k = a$$

$$\mathbf{v}_6 = \mathbf{u}_1 \mathbin{\%} k + b \cdot k + \mathbf{v}_3 \mathbin{/\!/} k \cdot k^2$$

$$\mathbf{q}_5 = r$$

For each transition $\delta(q, a) = (r, b, \mathrm{R})$:

$$\longrightarrow$$

$$\mathbf{u}_4 = b + \mathbf{u}_1 \cdot k \qquad \mathbf{q}_2 = q \qquad \mathbf{v}_3 \mathbin{\%} k = a$$

$$\mathbf{v}_6 = \mathbf{v}_3 \mathbin{/\!/} k$$

$$\mathbf{q}_5 = r$$

The sum-product of this FGG is 1 if $M$ halts on $w$, 0 otherwise. Therefore, computing the sum-product of an FGG is undecidable.

The operations $+, \cdot, \mathbin{/\!/}, \mathbin{\%}$ and $=$ can be further reduced to just the successor relation and equality with zero, as shown below.

$$\mathbf{x}_1 \;=\; \mathbf{x}_2 \;\rightarrow\; \mathbf{x}_1 \;\square\; \overset{\mathbf{x}_3 = \mathbf{x}_1 + 1}{(\mathbf{x}_3)} \;\square\; \overset{\mathbf{x}_3 = \mathbf{x}_2 + 1}{\mathbf{x}_2}$$

$$\mathbf{x}_1 \;>\; \mathbf{x}_2 \;\rightarrow\; \mathbf{x}_1 \;\square\; \overset{\mathbf{x}_1 = \mathbf{x}_3 + 1}{(\mathbf{x}_3)} \;+\; \begin{cases} \mathbf{x}_2 \\ \mathbf{x}_4 \end{cases}$$

$$(\mathbf{x}_1, \mathbf{x}_2) + \mathbf{x}_3 \;\rightarrow\; \mathbf{x}_1 = \mathbf{x}_3, \quad \mathbf{x}_2 = 0 \;\square\; \mathbf{x}_2$$

$$(\mathbf{x}_1, \mathbf{x}_2) + \mathbf{x}_3 \;\rightarrow\; \mathbf{x}_1 \ldots + \mathbf{x}_5 \;\square\; \mathbf{x}_3, \quad \mathbf{x}_3 = \mathbf{x}_5 + 1, \quad \mathbf{x}_2 \;\square\; \mathbf{x}_4, \quad \mathbf{x}_2 = \mathbf{x}_4 + 1$$

$$(\mathbf{x}_1, \mathbf{x}_2) \cdot \mathbf{x}_3 \;\rightarrow\; \mathbf{x}_1 \quad \square \; \mathbf{x}_2, \quad \mathbf{x}_2 = 0 \quad \mathbf{x}_3 \;\square, \quad \mathbf{x}_3 = 0$$

$$(\mathbf{x}_1, \mathbf{x}_2) \cdot \mathbf{x}_3 \;\rightarrow\; \mathbf{x}_1 \ldots \cdot \mathbf{x}_5 \,+\, \mathbf{x}_3, \quad \mathbf{x}_2 \;\square\; \mathbf{x}_4, \quad \mathbf{x}_2 = \mathbf{x}_4 + 1$$

Integer division and remainder can both be computed using the rule:

$$\begin{cases} \mathbf{x}_7 \\ \mathbf{x}_2 \end{cases} D \begin{cases} \mathbf{x}_1 \\ \mathbf{x}_3 \end{cases} \;\rightarrow\; \mathbf{x}_1 \cdot \mathbf{x}_5 + \mathbf{x}_7, \quad \mathbf{x}_2 > \mathbf{x}_3$$

where $\mathbf{x}_7$ is the dividend, $\mathbf{x}_2$ is the divisor, $\mathbf{x}_1$ is the quotient, and $\mathbf{x}_3$ is the remainder.