[Reviews · NeurIPS 2020]

Review 1

Summary and Contributions: The paper presents a new class of models, called Factor Graph Grammars. The paper shows that FGGs generalize a variety of existing classes of models, such as dynamic graphical models, SPNs, and so on.... The central idea uses hyperedge replacement grammars, which involves a set of production rules, terminal and non/terminal edges, similar to a context-free grammar. The production rules are applied onto the edges of the graph in the most direct sense, to generate graph fragments. The paper also discusses a conjunction operator (conjoining two FGGs that share some structure, e.g. nonterminal nodes), useful for limiting the derivations to only those that lead to some string. Finally, the paper analyzes the complexity of computing the partition function for FGGs. The complexity is dependent on the treewidth of the graph fragments from the production rules. There is also a way to expand out the FGG into a normal factor graph (sometimes impossible), and run standard inference techniques, although this has a worse complexity than the doing inference on the FGG directly.

Strengths: The subject of this work should be of interest for many people, and relevant to the community. Generalizing many existing graphical models can be fruitful in practice if inference algorithms for each specific case can be offloaded to a well-implemented inference algorithm for FGGs. There is also the possibility of more families of graphical models that can be discovered through this lens of FGGs, by considering different combinations of production rules.

Weaknesses: The requirements for the conjunction operator to be valid (Def 14) seems to be somewhat limited. It is not clear to me whether FGGs can be amenable to approximate inference algorithms, since in practice approximate algorithms need to be used for large graphical models. Line 226 seems to be relevant, but I'm not sure how this compares to approximate inference in some of the standard graphical models.

Correctness: Yes to the best of my judgment

Clarity: Yes

Relation to Prior Work: Yes

Reproducibility: Yes

Additional Feedback: Can you explain in more detail why Restricted Boltzmann machines cannot be represented using FGGs? The issue mentioned seems to be that it has two fully connected layers, but for example sum-product networks can also have somewhat fully connected layers in the sum layers (sum nodes can have more than two children), and there didn't seem to be trouble generating these structures in the Appendix. How does computing the partition function on the FGG versions of all these graphical models (plated factor graph, dynamic graphical models, case-factor diagrams, sum product networks...) compare to computing them directly on those graphical models? I have read the rebuttal and will keep my score as is.


Review 2

Summary and Contributions: The authors propose a new formalism for representing relation probabilistic models based on factor graphs. The formalism can accommodate repeated structures and tractable computation. Classes with tractable inference are identified.

Strengths: A new approach for obtaining expressive and tractable probabilistic models that subsumes several other formalisms.

Weaknesses: The paper is a overloaded with definitions and concepts, and there are not experiments, not even proof of concepts. More importantly, the authors do not make clear what are the advantages of the proposed approach compared to existing ones, including tractable relational models.

Correctness: The proofs are not in the paper; I haven't carefully checked the supplementary material.

Clarity: The text is well written, but the presentation could be improved by simplifying the terminology. For example, one could have external nodes as a particular type of labels (thus dispensing with def 5), refer to hypregraphs as bipartite graphs (thus dispensing with the oddity of allowing duplicate hype edges), etc.

Relation to Prior Work: The work relates to the vast literature of relational probabilistic models; in this sense the paper lacks scholarship. They also miss related work on tractable models with repeated structure (e.g. dynamic SPNs, tractable theorem proving)

Reproducibility: Yes

Additional Feedback: Consider a factor grammar that either "expands" a node into two new nodes, or "contracts" two nodes into one. That is, we have rules R1: v — [ X ] => v — [ ] — v1 — [X], v — [ ] — v2 — [X] R2 : v1 — [ X ], v2 — [ X ] => v1 — [ ] — v, v2 — [ ] — v I guess we can make that model into a finite language by constraining the recursion depth, similar to the grammar of the HHM for a given string shown in the paper. Now the model above would have few variables on the RHS graphs, but it would generate factor graphs with arbitrarily high tree width (so not amenable to sum-product inference). I can't see why such a model would be ruled out by Theorem 16. Small issues: - In line 190 it is written that sum-product is linear in the size; I guess it is meant to be exp in the size? - In line 219: what is a least solution?


Review 3

Summary and Contributions: Background: A graphical model such as a factor graph defines an unnormalized probability distribution over assignments (variable -> value mappings) that all have the same finite domain. New formalism: A factor graph grammar defines an unnormalized distribution over assignments that may have *different* finite domains. Technically, it's simply a hyperedge replacement grammar that generates a possibly infinite bag of factor graphs (which may share some variables and factors). Their size can be unbounded. The overall distribution is just the sum of the distributions specified by the individual factor graphs (so its support is the union of their supports). Algorithms: The authors describe how to take a product of two FGGs, which can be used to respectively describe the soft probabilities (constant across examples) and the hard evidence (different size and content across examples). This is similar to how a probabilistic relational model is "unrolled" against a database. The result is a new FGG. They then give an algorithm for finding an FGG's normalizing constant Z. Marginal conditional probabilities can standardly be found as by taking the ratio of two such normalizing constants, for two evidence-constrained FGGs that may be obtained by the product construction in the previous paragraph.

Strengths: This is an elegant formalism that is both expressive and remains computationally tractable. As the authors point out, it unifies some other formalisms. The key result is really that the normalizing constant Z of the FGG can be computed as the least solution of a quadratic system, just as for a PCFG (which is in fact a special case). It's aimed at unifying some different strands of work in the graphical modeling community behind a more powerful formalism. It is well-scoped, in that most of the important questions are answered (and neatly). I learned something from reading this paper, and can imagine making use of this formalism to explain or analyze more specialized families of models. Case-factor diagrams, which were a similar unifying attempt that is subsumed by this work, won best paper at UAI in the early 2000's (they are not so widely known today, but probably influenced PPLs).

Weaknesses: This is a theoretical paper that requires some background in graphical models and grammar-based methods to understand. So I'm not sure how much uptake it will get. But I don't think that's really a weakness. For the right audience, it should be quite clear. This kind of paper doesn't need experiments, but an efficient implementation would be nice (perhaps it could be added to the recently released Torch-Struct?).

Correctness: Definition 7 seems imprecise, since it doesn't talk about renaming nodes and edges from R to avoid duplicates. The examples show that this is done. But it's important to give the precise scheme, because in a set of factor graphs generated by the same FGG, we might want to know which nodes have the same name. That's important if we want to ask about the probability distribution "what's the probability that there's a variable X_3? conditioned on the X_3 existing, what's its marginal distribution? conditioned on X_3=0, what's the sum-product?" I'm also concerned that there may be spurious ambiguity in the derivations. If an intermediate graph has multiple nonterminals, can they be expanded in different orders to result in the same factor graph? That would mean that this graph is counted multiple times in the sum-product (see definition 10), which I think is unintentional. Shouldn't this be fixed by requiring derivations to follow a canonical expansion order, akin to leftmost derivation in a CFG? I suspect that theorem 18 on the undecidability of sum-product equalling 0 can be strengthened. Infinite domains make this problem undecidable even for the class of ordinary factor graphs, even if the domains are countable and the factors are restricted to be computable functions (Dreyer & Eisner 2009). This result holds even in the boolean semiring, i.e., it's really just a property of constraint programming. I think with finite domains, however, the problem is probably decidable by running your sum-product algorithm on the boolean semiring (which will make it terminate).

Clarity: The paper is fairly dense because of the unfortunate 8-page limit, but well and carefully written. I think the most confusing part for readers ls likely to be the conjunction operation -- if there's an extra page in the camera-ready, the presentation here should be slowed down with some qualitative discussion. (The mention of synchronous grammars is helpful, but in a synchronous grammar, rules are explicitly paired; in this case they are paired if conjoinable, which is .) You should probably clarify early on that you're talking about undirected hypergraphs. Notation in section 2.1: I regard 52-53 as a commutation property, basically vertices(\bar{e}) = \bar{vertices(e)}, where \bar lifts from variables or variable-tuples to their labels. I don't understand where the name "att" comes from ("attachment"?) or why you use the name "type" in the way you do. (If you have a function called "type", it should presumably map from tokens to types as \bar does, not from types to types.) (Ok, having read farther, I see that a node label is a data type, but an edge label determines not only the type of the potential function but also the specific potential function to use. But most of the previous comment still applies.) line 122 should be clearer that the PCFG is not directly defining a distribution over strings in a CFL, nor is it defining a distribution over derivation trees. Rather, it is defining a collection of factor graphs corresponding to different tree *topologies*. Each of these factor graphs defines an unnormalized distribution over derivations compatible with that topology, i.e., assignments of words and grammar nonterminals to the vertices of the factor graph. The overall distribution over derivation trees is then found by summing all of these unnormalized distributions! (Ok, now I see that you explain this pattern immediately under line 122, but not until after I'd followed line 122 to Example 19 in Appendix A and sorted this out for myself! And you don't tie the pattern back to PCFGs.) -------------------- UPDATE: Some reviewers did find the paper hard to follow, and we had a long discussion. Some of the big ideas had been too deeply buried. The paper is not easy to skim. While I advocate acceptance, I think you should rewrite carefully and more tutorially, particularly if you want the paper to have impact. Here are some things to make clearer: * Your FGG formalism is designed to combine expressiveness with tractability. (The expressiveness part came through on page 1; the tractability, not so much.) * PPLs are also expressive enough to deal with both the repeated-structure and the alternative-substructure problems ... but they are too general to support efficient inference methods. FGG has a claim to being about as expressive as you can get while still supporting tractable inference. If some part of a probabilistic program can be expressed or re-expressed as an FGG, then the procedure of Theorem 16 can be used to do inference on that part. * An FGG generates a possibly infinite set of factor graphs using a hyperedge replacement grammar. Each factor graph specifies an unnormalized distribution over structures, and the FGG distribution is a sum of all of these unnormalized distributions. (This kind of two-level setup was new to me -- is it in fact new?) * The advantage of using a hyperedge replacement grammar is that it allows the generation of factor graphs that include cycles (but still with bounded treewidth). The *derivation* is tree-structured, which is why inference by dynamic programming is possible, but the *derived structure* can have cycles. * What if you restricted to using a CFG instead of the hyperedge replacement grammar? Then the factor graphs generated by the CFG would all be tree-shaped (treewidth 1). I am not sure that restricting to a CFG would really limit expressivity, since a low-treewidth factor graph can be converted into a tree-shaped factor graph (junction tree). On the other hand, the original factor graph version has the advantage that it expresses stronger conditional independence assumptions than the junction tree. Those assumptions can be exploited during learning and *approximate* inference, even though they do not speed up exact inference beyond what you can get with the junction tree. * How would you get a PCFG from this? You would have to make two further restrictions. First, each variable in each factor graph would have to have domain of size 1, so that the trees you were generating were specific labeled trees and not distributions over trees. (This goes back to a one-level setup.) Second, the potential functions would have to be locally normalized. * Any factor graph can be generated by some FGG. But for any factor graph generated by a given FGG, the runtime of inference is exponential only in the length of the FGG rules (or, more tightly, in their complexity as characterized in Lemma 15). Specifically, there's always a good variable elimination order -- the reverse of the order in which the variables were introduced while deriving the factor graph. Thus, you can keep inference tractable by ensuring that your FGG's rules are short or simple. Moreover, there's a dynamic programming algorithm that can efficiently sum over *all* factor graphs generated by the FGG, as required by the FGG semantics -- basically the idea is to write a recurrence that sums over all the ways of nondeterministically expanding a given hyperedge. * Theorem 16 is the key tractability result. The construction given in the proof will work for any FGG -- you should have stated that in the theorem itself. However, the runtime bounds are only available for the cases 1. and 2. listed in the statement of the theorem. (Otherwise, you have to fall back on an iterative solution to a system of polynomial equations; how long that takes to converge numerically depends on the specific potential functions, so you don't make any statement about it.) The construction is closely related to computing the partition function of a weighted CFG (or even more relevantly, a weighted synchronous CFG, because that cannot always be binarized; cf. the cited Gildea 2011 paper). * The reason that you described things like HMMs and PCFGs was not because you handle those settings any better, but just to show that you subsumed other formalisms. (You would also support settings like Bilmes's GMTK for generalized factorial HMMs, where there are several random variables at each time step, and these interact in some sparse pattern with the random variables at the previous timestep.) Multiple reviewers wanted to see a realistic example that could not be handled with other formalisms short of PPLs. QUESTION: Above, I speculated the FGG formalism might be about as general as you can be while preserving tractability. (Or rather, tractability in arbitrary weight semirings. There are specific models that are not expressible with FGGs and yet are tractable thanks to special combinatorial algorithms: e.g., MAP inference for weighted bipartite matching, marginal inference for determinantal point processes, or either kind of inference for edge-weighted spanning trees. But those algorithms are semiring-specific.) Can this be made precise? I think you could show that your formalism generates all single MRFs with low treewidth. But that theorem wouldn't cover the cases where the FGG generates multiple MRFs.

Relation to Prior Work: If the idea is just to generate families of factor graphs, there are "probabilistic relational model" formalisms such as Markov logic that can also do that. There seem to be close connections that should be discussed. A PRM has to be combined with a database to produce a specific factor graph. (In general, the larger the database, the larger the resulting factor graph.) That step seems a lot like the "conjunction" method here. Furthermore, performing inference before joining (line 33) is analogous to "lifted inference" techniques in Markov logic. Is there anything that FGGs can do that PRMs can't? I know space is limited, but I would also like to see a little discussion of how FGGs relate to probabilistic programming languages. PPLs are the usual generalization of dynamic graphical models / PCFGs / case-factor diagrams (especially for the directed case). Only the conclusion mentions PPL; it suggests that FGG are powerful enough to handle a subset of PPL, but does either formalism subsume the other? What are the advantages and disadvantages? Kschischang et al. 2001 should probably be cited for factor graphs. Another connection (especially to Theorem 17) is the "Gates" paper by Tom Minka, which tries to augment graphical models with mixtures over structures, although not as flexibly as FGGs or PPLs. "Novel conjunction operation": this cross-product-of-nonterminals construction should be connected to database join, composition of string or tree transducers, and Cohen et al 2010, "Products of Weighted Logic Programs." (The connection to synchronous HRGs is already noted.) I wonder if "conjunction" is the best name, or if one could be chosen that better highlights these connections?

Reproducibility: Yes

Additional Feedback: line 120: actually, the vertices in Example 3 are numbered differently. I agree that an FGG can't be constructed that generates all restricted Boltzmann machines (if that's what line 136 means), but that would be a sum or mixture of many such machines, which is unusual (maybe you're thinking of https://arxiv.org/pdf/1001.0160.pdf ?). Also, I'd like to see the general principle about FGGs that rules this out. Is the issue the high treewidth? The fact that the number of edges grows quadratically with the number of nodes? The fact that there is no bound on the degree of a node? Would this also rule out topic models? (Note: stacked RBMs provide a further problem, namely that they are interpreted using conditionalization.) I think the term "finite FGG" should be avoided (e.g. line 244), as this suggests that there exist infinite FGGs in the sense of having an infinitely large specification. What you actually mean here is that the "FGL" L(G) is is finite (as you say in the section header). For theorem 17, it may be worth talking about where these finite cases tend to come from. Presumably the conjunction of a finite FGG with any other FGG is finite ...? (For example, an input string or lattice conjoined with an HMM or PCFG.)


Review 4

Summary and Contributions: The paper defines Factor Graph Grammars, which are a specialization of Hyperedge Replacement Graph Grammars (HRG) to factor graphs, a type of graphical model for probabilistic reasoning. The motivation is that factor graphs have fixed length, but several applications require variable-length models or parts of the model may have alternative substructures. They point out that several extensions of graphical models address these limitations, but that none addresses both at the same time, and that FGGs do. It then defines and analyses several theoretical aspects of FGGs, including inference methods, both direct and through conversion to regular factor graphs (in the faces in which the FGG defines a finite language). It finishes by pointing out that FGGs defining infinite languages involving infinite variable domains are undecidable. The paper is purely theoretical and there is no implementation or empirical evaluation of FGGs.

Strengths: The paper poses an interesting formalism and relates it to several other formalisms. Its first part is very well-written. It analyses several aspects of it, from definition to inference to computability. POST-REBUTTAL: after discussing with other reviewers, I see that the main novelty is a new *tractable* formalism. This is valuable, however the paper does not emphasize this, placing emphasis instead on variable-length and alternative-substructure. I strongly urge the authors to stress this aspect as well as to make more explicit and complete comparison to other tractable frameworks. Regarding my suggestion for drawing the graphs, which you mention you didn't see how it differs from you already doing, here's a reformulation. Currently, you have the "small black dots" on the LHS and the "black vertices" (external points) on the RHS. My suggestion is to get completely rid of the "small black dots" on the left (in fact you've already done so when the LHS is S) and to *move* the external points ("black vertices") from the RHS to the LHS. This is more natural because they are already present *before* the rule is applied. Notice how R2 independently drew an example FGG in their original review in the way I described, even before reading my review, which I believe lends evidence to its being a more natural way of doing it.

Weaknesses: - the paper is clearly and even elegantly written up to Section 5, when it becomes significantly more confusing and ambiguous. - it compares FGGs to other formalisms that do not address the two pointed limitations of regular factor graphs (variable length and alternative substructures), but does not mention the probabilistic programming formalism, which may address both limitations while being perhaps more standard and more direct to apply to problems. (It does mention probabilistic programming briefly in the conclusion, when it says that the authors have developed a simple probabilistic programming language to be translated into FGGs, but it is not clear to me why a translation instead of solving the probabilistic program more directly). - the lack of implementation, evaluation or at least an example of a problem in which it would be more useful than other frameworks. - regarding reproducibility, I think readers should be able to reproduce (understand) the theoretical results of the paper although it may require quite some work to go over some of the less clear points, as well as studying other papers in detail to understand the ideas that have been borrowed into this one.

Correctness: It seems correct although I found Section 5 not clear enough to really be able to tell.

Clarity: As mentioned above, it is quite clear until Section 5 and then not very much. Here are the main clarity points, in decreasing order of importance: Section 5: Line 190, "time linear in the size of the graph": isn't it exponential in the treewidth k? Line 192: The sentence right before Lemma 15 talks about reducing treewidth, but I don't see how either Lemma 15 or Theorem 16 help with reducing treewidth. Lemma 15 describes a process that creates a HRG with the same treewidth k as the original HRG, because if for every rule (L -> R) in G the treewidth of \bar{R} is at most k, then all right-hand sides have at most (k + 1) and the treewidth of G is already at most k. Then the new HGR has at most (k + 1) nodes in right-hand sides and therefore also treewidth equal to at most k. And Theorem 16 involves a single HRG, so there is a single treewidth involved and therefore no treewidth reduction. Line 219: what is a "least" solution? Line 268, “this ensures (…)”: not quite sure what this sentence means. Sections prior to 5: Line 130, "contain assignments for both": but what does it mean to "contain assignments for both"? If the disjoint union is a set of assignments, then it cannot contain the same assignment twice. If it is a multi-set, then it would be good to change the definition to include this fact. Another option is that the elements of the disjoint union somehow keep the derivation information too (it does sound like that is what you mean when you write "contains the assignments for both", which suggests each element in the set "remembers" which derivation it is *for*), in which case it would be better to define the disjoint union as a set of *pairs* of derivations and assignments. Line 131, w_G(\xi) = w_H(\xi): this seems to confirm that the disjoint set should be a set of *pairs* of derivations and assignments. Furthermore, the domain of w_G should also be of such pairs, otherwise w_G would not be a function at all since it might map the same assignment to more than one value. However, even if we assume w_G to be a function over pairs of derivations and assignments, I don't see how w_G defines a distribution over assignments (as stated it does), only over the aforementioned pairs. It seems that a distribution over assignments needs to be so that the potential for an assignment is the sum over all its possible derivations. If so, this needs to be stated. Line 181: this is not clear to me, particularly given the early confusion about whether the set being operated upon consists of assignments or (derivation, assignment) pairs.

Relation to Prior Work: Yes, although they leave out probabilistic programming. Perhaps they consider FGGs to be basically equivalent to probabilistic programs, but they seem like very different formalisms to me. POST-REBUTTAL: as mentioned above, please stress comparison to other tractable frameworks: cause-dependent factor graphs, PCFGs, tractable MLNs.

Reproducibility: Yes

Additional Feedback: Line 101: Drawing the left-hand side as a hyperedge (or, rather, as a hyperedge with a small black dot) as a reminder confused more than helped me. I found myself trying to remember if the small black dot was supposed to match some node. Also, hyperedges may contain no nodes, in which case drawing the small black dot is even misleading (and that is probably why you did not follow you own convention when you draw initial hyperedges S). Since in your factor graphs hyperedges are always drawn as squares and nothing else is drawn as a square, I think it would be clearer and more intuitive to draw the hyperedges on the left-hand side as simple squares. In fact, you already do what I suggest for initial hyperedges, as I mentioned before, in the first production rule drawn under Example 9 as well as for all left-hand sides in Figure 2 and it is perfectly understandable and intuitive. PS: actually, after some more thought, I would suggest a further improvement: to draw the external points on the *left*-hand side of the production rule, connected to the hyperedge being replaced. This would give an intuitive idea of the sort of pattern being matched for replacement. If needed, the external point may appear again on the righ-hand side, making it more clear how the substituted nodes connect to the already existing ones. This scheme would also prevent the non-intuitive presence of disconnected external points like T1 in the production rules of Figure 2, because in that case it would appear only on the left-hand side, more clearly indicating its sole function as a matching point. Line 107, “copy”: It might be good to remark that this copy changes the "name" of the vertices so they are unique in the destination graph. For example, in the last transition in Example 9 T_2 became T_4. This is especially important since these names are not just arbitrary tags (analogous to object memory positions in an implementation), but are actually significant when used for establishing a correspondence between nodes in G and G_w when Conjunction is presented in Section 4. Line 108: this sentence takes a while to understand because "for each i" raises the question "i in *what*?" Perhaps "For each ith endpoint v of e, identify v with (...)" would be better. Line 224: it might be worth mentioning right away at case 2 that it applies only to cyclic cases, as opposed to doing so in line 228. Line 231: Why don't cases (1) and (2) always apply even if the condition in line 230 is not satisfied? Line 256: definition of CondOne is confusing. So far you had presented B as an array of booleans, but here you are treating it as an individual boolean variable. You should use a different letter for the arguments of CondOne when defining it generally. It would also be better to introduce it after its first use instead of in advance. The same observations hold for Cond below. Minor: Line 141: “inside and Viterbi” -> “inside-out and Viterbi” Line 180: I was not familiar with "Viterbi semiring". I see now it is the semiring for maximization. A reference or short hint may help similar readers. Line 204: probably better to add "for any non-terminal X".

[Author Response · NeurIPS 2020]

We thank all four reviewers for their thoughtful reviews, and are happy that they value the contribution of a new
expressive, unifying formalism that maintains tractable inference. We acknowledge that this is a purely theoretical
paper; an implementation is on our agenda, but we felt that there was already more than enough here to write up.
Reviewer 4 (**R4**) asks for "at least an example of a problem in which it would be more useful than other frameworks."
We offer, if we may, a recent exchange on Twitter,[1] where Charles Sutton asked whether there was a procedure that
could go from a description of a PCFG to an efficient inference algorithm for trees given strings. In our opinion, no
satisfactory answer was given, but FGGs provide such a procedure, sketched in Appendix A. We assumed that the CFG
is in Chomsky normal form, but Lemma 15 would automate that as well.

**Formalism**  **R2** gives an example FGG that appears to generate graphs with arbitrarily high treewidth. If we understand
the notation correctly, Rule R2 has two hyperedges on its left-hand side, which is not allowed by the definition of HRG.

**R4** provides feedback about our notation for left-hand sides, which are drawn as a hyperedge together with the same
number of external nodes as the right-hand side, to show the pattern that gets replaced with the right-hand side; sorry if
this wasn't clear. We are uncertain how **R4**'s suggestion to draw the external nodes on the left-hand side differs from
what we already do. (Except Figure 2 – thanks for drawing our attention to the missing external nodes there.)

**Renaming**  We agree with **R3** and **R4** that Def. 7 should formalize how nodes are renamed when copied. We'll fix it,
but we'd also like to clarify two points. First, **R3** writes that we want to know which nodes have the same name so
we can query by name. But this wouldn't be as flexible as we'd like; for example, we'd like to query a HMM for the
probability that the last tag is N, but the last tag has different names in different graphs. Enabling such queries is why
we developed conjunction. Second, **R4** writes that we need node names when matching up nodes during conjunction.
But conjunction is an operation on FGGs, not factor graphs, so at the time of conjunction, no renaming has taken place.

**Ambiguity**  **R3** asks whether there is spurious ambiguity due to the ordering of rewrites. We are assuming (and can
make explicit) that a derivation is a tree, not a sequence of rewrites, so we don't count different orderings separately
in the sum-product. However, **R4** asks about different derivation *trees* yielding the same graph, and we do treat these
separately in the disjoint union and count them separately in the sum-product, so indeed, a FGG defines a distribution
over (derivation, assignment) pairs. We agree that the notation should be improved and will think about how to do so.

**Exact inference**  **R2** and **R4** ask about the claim (line 190) that sum-product is linear in the size of the graph. It's linear
in graph size and exponential in the treewidth $k$, but $k$ is a property of the grammar, independent of graph size.

**R4** asks about the claim on line 192 that Lemma 15 reduces treewidth; we apologize that this is indeed not worded
correctly. Lemma 15 does not change the generated graphs and cannot change their treewidth. Rather, Theorem 16's
time complexity depends on the size of the right-hand sides, and Lemma 15 reduces the size of the right-hand sides.

**R4** asks why case (1) and/or (2) at lines 223–5 are not applicable if the condition at line 230 is not satisfied. If $X$ can
derive a graph with two occurrences of $X$, then we can't solve the resulting system of equations by substitution because
of the circularity, and we can't solve it using linear algebra because some of the equations are quadratic.

**Approximate inference**  **R1** asks whether approximate inference can be done on FGGs. This was one motivation behind
§5.2, since converting a finite FGG to a factor graph makes it possible to use any inference algorithm. Approximate
inference on infinite FGGs remains an open question, which we consider an important direction for future research.

**RBMs**  **R1** and **R3** ask to clarify our claim that FGGs can't generate RBMs. We mean that an FGG can't generate the
infinite set of all RBMs; this example comes from Obermeyer et al. (2019) and would be of interest in a situation where
one wants an FGG that can be conjoined with another to yield a single RBM. Since an $m \times n$ RBM has treewidth
$\min(m, n)$, the set of all RBMs has unbounded treewidth and can't be generated by a FGG.

**Other formalisms**  **R1** asks how the complexity of inference for FGGs compares to the complexity of inference in the
formalisms we compare them to. We think the complexity should be the same in the cases considered in Appendix B
and can expand the proofs to include complexity analyses.

**R2** asks for comparison with PRMs, **R4** for comparison with PPLs, and **R3** for both. Our comparisons were focused
on more constrained formalisms, but we'll be happy to discuss these more general formalisms as space permits. We
suspect both PRMs and PPLs can simulate any FGG, but doubt whether the simulation would allow exact inference
with the same efficiency. **R2** mentions tractable relational models, and we'd be interested in looking further into any
citations that **R2** might provide.

**Thanks**  We thank the reviewers for providing other missing citations, especially **R3**, and we are grateful for the many
other comments from all the reviewers, which we will address in the final version of the paper.

## Footnotes

[1] `https://twitter.com/RandomlyWalking/status/1258307220371984384`


[Meta-Review · NeurIPS 2020]

This paper presents an interesting formalism, but it needs to better clarify its limitations, its possibilities, and its relationships to other formalisms. I strongly encourage the authors to make use of the detailed feedback in the reviews so that this paper lives up to its potential -- the reviewers went above and beyond in discussing this paper at length, much more than any other paper I saw this year.